# GO-TiO_2_ as a Highly Performant Photocatalyst Maximized by Proper Parameters Selection

**DOI:** 10.3390/ijerph191911874

**Published:** 2022-09-20

**Authors:** Aida M. Díez, Marta Pazos, M. Ángeles Sanromán, Yury V. Kolen’ko

**Affiliations:** 1Nanochemistry Research Group, International Iberian Nanotechnology Laboratory, Avenida Mestre José Veiga s/n, 4715-330 Braga, Portugal; 2CINTECX, Grupo de Bioingeniería y Procesos Sostenibles, Departamento de Ingeniería Química, Campus Lagoas-Marcosende, Universidade de Vigo, 36310 Vigo, Spain

**Keywords:** titanium dioxide doping, graphene oxide, methylthioninium chloride, electrochemical characterization measurements, photodegradation, process and catalyst characterization

## Abstract

The synthesis and characterization of novel graphene oxide coupled to TiO_2_ (GO-TiO_2_) was carried out in order to better understand the performance of this photocatalyst, when compared to well-known TiO_2_ (P25) from Degussa. Thus, its physical-chemical characterization (FTIR, XRD, N_2_ isotherms and electrochemical measurements) describes high porosity, suitable charge and high electron mobility, which enhance pollutant degradation. In addition, the importance of the reactor set up was highlighted, testing the effect of both the irradiated area and distance between lamp and bulb solution. Under optimal conditions, the model drug methylthioninium chloride (MC) was degraded and several parameters were assessed, such as the water matrix and the catalyst reutilization, a possibility given the addition of H_2_O_2_. The results in terms of energy consumption compete with those attained for the treatment of this model pollutant, opening a path for further research.

## 1. Introduction

Photocatalysis has been regarded lately as a wastewater treatment alternative. It has several advantages, namely the non-selective degradation of pollutants and the possibility of working with sunlight [1].

Nevertheless, in order to achieve acceptable results when compared to other alternatives, such as absorption, researchers are focused on UVA-photocatalysis, where high degradation rates can be achieved by the utilization of the high performant TiO_2_ [2].

However, the activity of TiO_2_ seems to be not enough for scale-up applications, as little output has been found in real applications. This lack of real usability can be explained by the drawbacks of TiO_2_, such as its activation only under UVA radiation [3] and its insufficient absorption capacities [1].

In order to overcome this, some modifications have been performed to commercial TiO_2_ (P25 from Degussa), such as doping with elements to reduce its band gap, which enhances its activation under wavelengths higher than UVA. For instance, Chakinala et al. [3] added Bi and Ag to TiO_2_ to more efficiently degrade contaminants such as methylthioninium chloride (MC) or rhodamine B. Likewise, the addition of porous compounds, such as SiO_2_, has been essayed, favoring a reduction in the agglomeration between photocatalyst particles and promoting absorption due to an increase in photocatalyst global porosity [1]. In this regard, other authors have proposed the addition of GO [4,5], although the efficiencies attained can be widely improved with more sophisticated reactor set-ups.

With these alternatives for efficiency enhancement, photocatalysis processes could compete with other less expensive alternatives, such as absorption, which has to cope with spent absorbent disposal. Another approach that facilitates the real application of photo-based processes is to synthesize photocatalysts with dual activity under UVA and visible radiation [3,5]. With this, the treatment times are not subject to either sunlight or lamp operability.

Nevertheless, few researchers [1,3,6] have focused on the synthesis of dual activity photocatalysts, although increasing attention on the topic has been noticed (Appendix A). Moreover, in those cases, the catalysts were expensive, either due to the presence of costly elements, such as Ag [3], or due to the high temperature increase requirement in the synthesis processes [1,6].

This study proposes the utilization of GO-TiO_2_ as a dual activity photocatalyst for the degradation of a model drug (MC). For this, research was carried out under UVA radiation and validated afterwards with simulated solar radiation. Moreover, the photocatalyst was characterized by traditional measurements, such as X-ray diffraction (XRD), Fourier transform infrared spectroscopy (FTIR), N_2_ isotherm and bang gap calculation, and by poorly studied electrochemical measurements, such as impedance or electrochemical active surface area (ECSA). The unstudied photocatalyst reusability was also assessed.

## 2. Materials and Methods

### 2.1. Reagents

MC of high purity was purchased from Alfa Aesar. GO (>99%) was bought from ACS Material. H_3_BO_3_ (>99.5%), (NH_4_)_2_TiF_6_ (>99.9%) and ethanol (>99.8%) were acquired from Sigma. TiO_2_ was acquired from Evonik (so-called P25), which contains 77.1% anatase, 15.9% rutile and 7% of amorphous TiO_2_. Milli-Q water was attained from an Advantage A10 system (Millipore).

### 2.2. Catalyst Synthesis

The catalyst was synthesized using a previously reported procedure for the treatment of industrial dyes [7], so that this study can deeply evaluate the photocatalyst performance with the study of the drug MC. Briefly, it consists of the dispersion of GO (10 mg/mL) in 10 mL of ethanol. Then, 0.1 M of (NH_4_)_2_TiF_6_ is added dropwise under stirring, as well as H_3_BO_3_ 0.3 M. This was kept for 1 h under vigorous stirring. Then, the prepared mixture is placed on a hydrothermal autoclave at 60 °C during 2 h. Then, the product is placed on the furnace at 200 °C for 2 h so 1GO-TiO_2_ can be attained. The dosage of GO was studied and consequently, ½GO-TiO_2_ and 3-GO-TiO_2_ were synthesized.

### 2.3. Catalyst Characterization

The characterization of TiO_2_ was compared to the best performant GO-TiO_2_ catalyst for MC photodegradation. Room-temperature XRD was carried out on a Rigaku diffractometer with an X PERT PRO MRD (Pananalytical, USA), using Cu-Kα radiation (λ = 1.54187 Å). Each pattern was recorded in the 2θ range from 20 to 80° with a step of 0.03° and the total collection time of 2 h. The analysis of XRD patterns was carried out with Higscore software 3.0 (Pananalytical, Westborough, MA, USA).

FTIR measurements were performed using the NBI—Bruker Vertex 80 v vacuum FTIR of Bruker within 400 and 4000 nm under N_2_ conditions.

The UV–Vis spectra of the photocatalyst were measured in a 1 wt % ethanol matrix using a spectrophotometer (V-2550, Shimadzu, Kyoto, Japan).

The N_2_ isotherm was carried out with Autosorb IQ2 (quantachrome) by degassing the sample at 303 K for 4.5 h and introducing N_2_ at 77.35 K.

The point of zero charge (PZC) was evaluated by plotting the set pH (with HCl or NaOH) vs. the difference in pH after 24 h stirring, following reported procedures [5].

Electrochemical impedance spectroscopy (EIS) and ECSA were measured using the Autolab PGSTAT302N of Metrohm (Herisau, Switzerland). A three-electrode system was used, with Pt as the counter electrode, calomel electrode as the reference and Ni-foam (1 cm^2^) as the working electrode, where the catalysts were impregnated. Then, 3 mg of the catalysts were dissolved in 630 μL of a mixture of Nafion:EtOH (1:20). The experiments were carried out in a 0.5 M Na_2_SO_4_ solution. EIS was measured on a frequency range of 0.1–10^5^ Hz with a sinusoidal perturbation of 10 mV. ECSA was measured by measuring cyclic voltammetries (CVs) in a range of 0.1 V, so it crossed the 0 current. These CVs were measured at scan rates between 10 and 200 mV s^−1^. Then, a linear trend was obtained by plotting the scan rate vs. half the difference in current density between the anodic and cathodic sweeps. The linear fitting slope of this graph provides the geometric double-layer capacitance (CDL). ECSA is calculated following equation 1, where Cs is the specific capacitance and has a value of 40 μF cm^−2^ [8].
ECSA = CDL/Cs(1)

### 2.4. Reactor Set-Up

Taking into account the porosity provided by GO addition, the equilibrium under stirring on dark conditions was maintained for 30 min. The photocatalytic performance was evaluated on two reactors. (I) The wide-open-reactor was cylindrical, with a diameter of 10 cm and a height of 1.5 cm, in order to promote photocatalyst activation by the lamp placed at 2 cm above (Figure 1), although a 6 cm gap was also assessed. The UVA-LED lamp used was acquired from Seoulviosys model 78 CMF-AR-A03 (365 nm, 4.8 W). (II) In order to test the efficiency of the high radiated surface area, a test that switched the wide-open reactor for a narrow-open reactor was carried out (Figure 1B). This reactor has a 5 cm diameter and is 10 cm high. (III) A test with simulated solar radiation was carried out with a 400–740 nm emitting lamp (600 W, from Toplanet) placed at 2 cm next to the narrow-open reactor, so a high surface area was irradiated (providing that glass is transparent to visible radiation) (Figure 1C).

All preliminary experiments were carried out using Milli-Q water (Sartorious) as the working matrix, with a MC concentration of 20 mg/L. However, validation tests were performed with real wastewater, kindly donated by WTP (wastewater treatment plant) of Guillarei, Tui, Galicia, which remediates the municipal effluents by a primary process (so-called physically treated water) and subsequent secondary treatment (physically + biologically treated water). The characteristics of those effluents are provided in the Appendix A).

### 2.5. Process Efficiency Analysis

Decoloration of the samples was followed using Equation (2), where A_i_ is the initial absorbance and A_t_ is the absorbance at a time t. Absorbance was measured using the spectrophotometer (V-2550, Shimadzu, Kyoto, Japan). A calibration curve was produced in order to assess the exact remaining MC concentration with time.
Decoloration = (A_i_ – A_t_/A_i_) × 100(2)

Energy consumption (EC) was measured in order to evaluate the efficiency of the processes and to compare the results. For that, Equation (3) was used, where L is the lamp power (W), t is the treatment time (h) and C is the amount of degraded MC (mg).
EC (Wh/mg) = L × t/C(3)

## 3. Results

### 3.1. Preliminary Tests

#### 3.1.1. GO Dosage

Initially, in order to select the catalyst to work with, the influence of GO dosage was studied, as the amount of porous agents added to TiO_2_ has been shown to have an effect on photo-activity [1,4]. In fact, some differences regarding dark absorption and photodegradation were detected (Figure 2). The MC degradation enhancement is remarkable when comparing well-known TiO_2_ to TiO_2_ with any GO dosage. Consequently, the synthesized 1GO-TiO_2_ was prepared with half (½GO-TiO_2_) and three-fold (3GO-TiO_2_) the amount of GO. In addition, ½GO-TiO_2_ was the best candidate in terms of photocatalysis performance, and it was thereafter called GO-TiO_2_. 

#### 3.1.2. Reactor Set-Up

In order to confirm the suitability of the proposed reactor (Figure 1A), the comparison between the aforementioned reactors was carried out. The results provided in Figure 3 demonstrate the importance of irradiation surface and intensity. In fact, EC was calculated for both the wide-open-reactor, with the lamp placed at 2 and 6 cm, and for the narrow-open-reactor and provided EC values of 0.12, 0.21 and 0.18 Wh/mg, respectively. Consequently, the open-wide-reactor with the lamp placed at 2 cm was used thereafter.

### 3.2. Catalyst Characterization

#### 3.2.1. XRD

The XRD data of both TiO_2_ and GO-TiO_2_ demonstrate that the former is stable in the synthesis process (Appendix A), possibly due to the mild conditions during the GO addition.

#### 3.2.2. FTIR

The FTIR spectra (Appendix A) show that GO-TiO_2_ has characteristic peaks of both GO and TiO_2_, demonstrating successful synthesis.

#### 3.2.3. N_2_ Isotherms

Figure 4a depicts the N_2_ isotherm of both photocatalysts that fit a III type isotherm, which indicates the presence of macropores and a weak interaction photocatalyst-MC [9]. GO-TiO_2_ showed significantly higher absorbance of N_2,_ which may be related to its higher absorption capacity. Surface area, pore diameter and volume are depicted in Appendix A.

#### 3.2.4. PZC

The PZC was calculated, and some differences were detected between TiO_2_ and GO-TiO_2_ (Figure 4b). Indeed, the PZC was more than 1 pH unit of acid than the commercial TiO_2_. The TiO_2_ PZC is in concordance with other studies [10].

#### 3.2.5. UV Spectra

Both catalyst dispersions in ethanol (1% wt) were measured between 190 and 900 nm and the resulted profile is shown in Appendix A, where a slight augmentation in the absorbance response can be observed, which can be related to higher photo-activity [1]. Tauc plots were calculated following previous protocols [1,2] (Appendix A), and band gap values were measured, being 3.2 and 2.2 eV for TiO_2_ and GO-TiO_2_, respectively.

#### 3.2.6. Electrochemical Measurements

The relationship between photo-activity and electrochemical activity has been demonstrated previously [11,12]. Consequently, the electrochemical impedance spectroscopy with the attainment of Nyquist graphs, and the calculation of the ECSA were carried out (Figure 5), in which the data can be fitted to equivalent circuits (Appendix A). The series resistance represents the electrode, which is practically constant for both catalysts, considering the usage of the same set-up. In the case of the parallel resistance, it is related to the layer resistance of the material, with a 4-fold smaller resistance in the case of GO-TiO_2_ when compared to TiO_2_.

Regarding the Nyquist graphs (Figure 5a), the semicircle arc is smaller in the case of GO-TiO_2_. It can be also noticed, when comparing CVs from TiO_2_ and GO-TiO_2_ (Figure 5b,c, respectively), that the broadening of the CVs is practically twice in the case of GO-TiO_2_ (Figure 5c), obtaining a range of 0.2 mA compared to the 0.11 mA for TiO_2_ at 200 mV s^−1^. 

Regarding the CDL results, GO-TiO_2_ provided a more than double CDL value, demonstrating that the increase in electroactive surface area has been accomplished, which is in concordance with the N_2_ isotherm results (Figure 4a).

### 3.3. MC Degradation

#### 3.3.1. Duality UVA–Vis

To our knowledge, the number of studies centered on dual activity UVA–Vis is much smaller than those on the independent evaluation of the radiation source (Appendix A). Although, the trend is increasing with time, due to the obvious advantage of the possibility of working with daylight and with low consumption lamps during the night. Consequently, the synthesis and study of dual activity photocatalysts should be carried out in order to facilitate real usages of photodegradation processes.

The photo activity of GO-TiO_2_ turned out to be higher than TiO_2_ (Figure 6); thus, GO-TiO_2_ favors MC degradation not only under UVA, but also under visible radiation when compared to TiO_2_.

#### 3.3.2. Catalyst Reuse

Catalyst reuse directly affects the process cost [3]; thus, it was evaluated. Lower MC degradation rates were reported (Figure 7a) for the 1st cycle (100%) to 2nd (70%) and 3rd cycles (38%). As a solution, H_2_O_2_ (0.66 mg/mL) was added, as some references have tested the enhancement of TiO_2_ photocatalyst performance with H_2_O_2_ addition [13]. In this case, the efficiency was kept constant during the three cycles (Figure 7b).

#### 3.3.3. Matrix Effect

The working matrix is known to have an impact on photocatalysis degradation [14,15]; thus, its effect was validated. In fact, Figure 8 demonstrates that it was not only the degradation that diminished when using real effluents, but also it was a slight reduction in the absorption.. Indeed, a degradation decrease of 27% in the case of the physical and biologically treated effluent and of 38% in the case of the physically treated effluent was observed, when compared to the MC treatment with Milli-Q water.

## 4. Discussion

### 4.1. Preliminary Tests

#### 4.1.1. GO Dosage

Slight differences in MC degradation were detected when the ratio of GO varied from 1/2 to 3 (Figure 2), although variations were noticed on the adsorption behavior (at time 0). The higher the GO dosage, the higher initial absorption in dark conditions, which is in concordance with previous studies and reflects the porous characteristics of GO [4,5]. However, the photocatalytic behavior dramatically changed; thus, 3GO-TiO_2_ caused 67% absorption in darkness, increasing only to 83% MC degradation after 30 min of illumination. On the other hand, ½GO-TiO_2_ caused 15% absorption and MC photodegradation reached 100% after illumination. This demonstrated that the lower the GO quantity, the lower the absorption within the GO porous structure and the better the photocatalytic activity. This also demonstrates the fact that too much MC absorption within the photocatalyst structure hinders the photocatalytic activity, due to the blockage of active sites [5], which also explains the decrease during the catalyst reuse process, due to the adsorption of MC on the photocatalyst (Figure 7a).

Additionally, ½GO-TiO_2_ (called from now on GO-TiO_2_) surpasses commercial TiO_2_ performance by 30%. This may be caused by the enhancement of porosity and hydrophobicity caused by GO addition [5]. These results are in concordance with the work of Mahanta et al. [1], who noticed that combining TiO_2_ with porous SiO_2_ enhanced TiO_2_ activity, but the lowest concentration was the most optimal. Additionally, GO-TiO_2_ improvement may be caused by easier photocatalyst activation, given the smaller band gap of GO-TiO_2_ (Appendix A).

#### 4.1.2. Reactor Set-Up

The proposed wide-open reactor has a high surface area (10 cm) and the radiation source is placed extremely close to the bulb solution (2 cm), which enhances the photodegradation process. Indeed, this statement was corroborated by the results depicted in Figure 3. Thus, when placing the open-wide reactor at 6 cm instead to 2 cm, the degradation performance was diminished by more than 40%. This is in concordance with other studies, which have highlighted the decrease in the light intensity with distance. For instance, Anku et al. [10] accomplished 95% methyl orange degradation with a xenon lamp placed at 10 cm, whereas only 55% was achieved when the lamp was placed at 13 cm. This is in concordance with the work of Casado et al. [16], who reported that the radiation W that reached the reactor surface reduced with distance. For instance, when placing an LED lamp at 4 cm from the solution, 15 W was reported. In contrast, if they placed the lamp at 8 cm, only 5 W was recorded. Moreover, MC degradation suffered a decrease of 30% when reducing the illuminated surface area from 10 to 5 cm (when changing the open-wide reactor to the narrow-open reactor). In this case, this is in concordance with the intensity of radiation by the illuminated surface, and although the intensity is higher on the very center, extra photons can reach the photocatalyst [16] in the case of the open-wide reactor.

In both cases, the decrease is caused by the reduction in light intensity, reducing the number of photons that can reach the surface of the catalyst [10]. When focusing on the degradation behavior with time, one can notice that until the first 10 min, the performance of MC degradation by the narrow-open reactor is worse than with the open-wide-reactor placed at 6 cm; this may be caused by the light scattering of the concentrated MC solution and the depth of the effluent in the narrow-open reactor. As time passes, the darkness of the solution is diminished and with that, light scattering is reduced. Consequently, after 10 min of reaction, the MC degradation performance is higher with the narrow-open reactor placed at 2 cm than with the open-wide reactor placed at 6 cm.

### 4.2. Characterization

#### 4.2.1. XRD

Regarding XRD, the samples showed almost the same profile (Appendix A), where anatase and rutile phases are depicted. This is concordance with other authors, which noticed that TiO_2_ was stable during superficial doping, demonstrating the high stability of this material. For instance, Kurniawan et al. [4] added from 0.05 to 50% of GO to TiO_2_ in order to enhance MC degradation and the XRD did not significantly change, as GO has low crystallinity.

#### 4.2.2. FTIR

FTIR measurements (Appendix A) demonstrate the broadening of the peaks due to the bonds generated within GO-TiO_2_. This is in concordance with the work of Kurniawan et al. [4], who labelled the peak at around 1400 cm^−1^ as a H-O typical of TiO_2_ samples and reported that the new peaks between 1600 and 1700 cm^−1^ were caused by C = C and C = O bonds, which are found in the GO sample. Indeed, Appendix A demonstrates the presence of several functional groups, which are known to promote worthy photocatalytic activity [5], promoting charge transfer.

#### 4.2.3. N_2_ Isotherms

Both TiO_2_ and GO-TiO_2_ showed type III isotherms related to the presence of macropores [9]. GO-TiO_2_ showed higher N_2_ absorption (Figure 4a), which is related to a 19% surface area increase in GO-TiO_2_ when compared to TiO_2_ (Appendix A), for which the pore diameter and volume are higher due to surface clogging with GO and the increase in the particle due to process synthesis [9,17].

These results are in concordance with the fact that GO-TiO_2_ showed a 15% absorption rate (Figure 2); thus, this absorption may be caused by the physical trapping of MC due to the higher surface area of the synthesized photocatalyst. Indeed, other authors have experienced higher MC absorption on their photocatalysts when the surface area was higher [18].

#### 4.2.4. PZC

The PZC is a way of measuring the possible electrostatic interactions between MC and the photocatalysts. Thus, the absorption of MC within GO-TiO_2_ may be caused not only by the physical trapping due to the higher surface area but also due to charge attraction. 

Indeed, the PZC of TiO_2_ is 6.55 and PZC of GO-TiO_2_ is 5.19 (Figure 4b); this means that at a pH below this value, the absorbent would be positively charged and at a pH above the PZC, the absorbent would be negatively charged.

Thus, considering the MC structure where the large organic part is positive, negatively charged absorbents would attract more of this pollutant. Considering that the PZC of GO-TiO_2_ is lower than TiO_2_, the aforementioned negative charge is more common in the former and with MC interaction. In fact, the pH of the MC 20 mg/L solution is 6.67, so it is clear that TiO_2_ would almost not be negatively charged (PZC = 6.55), whereas GO-TiO_2_ would be negatively charged, as there is a 1.5 pH unit difference. Thus, the interaction between MC-GO-TiO_2_ is not only physical, but also electrostatic.

#### 4.2.5. UV Spectra

The UV–Vis response of GO-TiO_2_ was evaluated and compared to that of TiO_2_ and the results (Appendix A) demonstrate the much better activity of the synthesized GO-TiO_2_. Thus, GO-TiO_2_, with a band gap of 2.2 eV, would be more easily activated under radiation than TiO_2_ (3.2 eV, which is in concordance with previously reported data) [1,2]. Moreover, the GO-TiO_2_ band gap means that it is also activated under visible radiation (Section 3.3.1), due to the smaller electron–hole recombination rate [4].

#### 4.2.6. Electrochemical Measurements

The Nyquist graph (Figure 5a) reflects a higher electron mobility in the case of GO-TiO_2,_ as it presents a smaller arc radius [2]. This higher electron mobility has been highlighted as a reason for photocatalyst performance improvement in the GO-TiO_2_ samples [5]. Thus, GO addition favors the movement of electrons and foments the charge carrier transference from TiO_2_ to GO, avoiding electron–hole recombination. This fact is demonstrated by the better performance of GO-TiO_2_ when compared to TiO_2_ (Figure 2). Indeed, the smaller value of the parallel resistance (Appendix A) (17.5 kΩ vs 75.2 kΩ) indicates a lower charge-transfer resistance, which enhances the photocatalytic activity [2].

Moreover, increasing the surface area (BET) (Appendix A) and ECSA (Figure 5d) also has a positive effect on the degradation performance, due to the increase in the surface reactions of photogenerated carriers [5].

### 4.3. MC Degradation

#### 4.3.1. Duality UVA–Vis

The results provided in Figure 6 demonstrated the activity of the GO-TiO_2_ photocatalyst under visible radiation. This is explained by the smaller band gap (from 3.2 of TiO_2_ to 2.2 eV of GO-TiO_2_) when adding GO to TiO_2_ [4]. Thus, even though this photocatalyst adsorbs 15% more MC than TiO_2_, the degradation performance is doubled with this synthesized catalyst. Other authors have demonstrated the dual UV and visible activity of their synthesized catalysts, such as V/Mo-TiO_2_ [6] or TiO_2_-SiO_2_ [1], although GO-TiO_2_ may be an easier-to-apply option. For instance, Chakinala et al. [3] had already demonstrated the duality of a catalyst that was useful for both UV and visible radiation. Nevertheless, their catalyst was more complex and expensive (Bi/Ag-TiO_2_). Other authors have reported that TiO_2_ can experience a shift to visible radiation with the addition of compounds such as GO [1].

Nevertheless, regarding the UV–Vis spectra, no significant shift was detected as in the case of Mahanta et al. [1] for their TiO_2_-SiO_2_ photocatalyst. These authors explained the dual activity of UVA-visible radiation by the small fraction of wavelengths in the visible range, which are enough to activate the photocatalyst due to SiO_2_ addition in their case, or GO in this study.

#### 4.3.2. Catalyst Reuse

The efficiency of the photolysis process is diminished after the second cycle (Figure 7a). This is caused by the blocking of the active sites with MC molecules and also by the generation of more stable intermediates [3]. In both cases, the addition of an extra oxidant, such as H_2_O_2_, is a good alternative (Figure 7b), by helping with the cleaning of the active sites and favoring the degradation of stable intermediates. In fact, the combination of TiO_2_ and H_2_O_2_ promotes the generation of peroxy-titanium complexes (-TiOOH), which promote the generation of HO· radicals [13]. Moreover, extra HO· are generated by the H_2_O_2_ reaction with the electrons from the conduction band of GO-TiO_2_ and also with the superoxide radicals that are produced by the photodegradation process [19].

The presence of macropores (Section 4.2.3) makes desorption difficult and this can explain the fact that in the photocatalyst reuse process, the second and third cycles did not show absorption because the MC within the structure was not degraded during the first cycle (Figure 7a), demonstrated by the blueish color of the photocatalyst. In the case of H_2_O_2_ addition the oxidants generated are enough to degrade the MC absorbed, a fact demonstrated by the clean photocatalyst after reuse and the restored 15% adsorption between cycles, as shown in Figure 7b. These results are superior to previous data, where the performance of the catalyst was also reduced over the cycles [1,3]. These authors could have considered H_2_O_2_ addition in order to reuse the photocatalyst. This opens a path for real usages, where photocatalyst reuse could reduce waste disposal and simplify plant operation.

#### 4.3.3. Matrix Effect

Figure 8 demonstrates the negative effect of the usage of real wastewater as matrixes. This is caused by the presence of organic and inorganic matter in those effluents (Appendix A). Those compounds can act as scavengers and reduce the rate of the degradation process [14]. Moreover, even absorption under dark conditions was slightly reduced; this can be caused by the absorption of the other organic matter present in the real effluents, which cause an increase in COD (Appendix A), and by the clogging of the active pores, due to the presence of suspended solids. Indeed, other authors [20] have noticed a reduction in compound absorption under dark conditions with salts such as NaCl or NaHCO_3_, or when treating real water, such as river water and treated wastewater. This can be explained by a modification in the electrostatic attraction between the photocatalyst and the contaminant due to the salt content variation [20]. Thus, in real effluents, the ionic strength and the organic matter content vary, leading to particle agglomeration [15].

#### 4.3.4. Comparison with Previous Research

The achieved results demonstrate the importance of suitable photocatalyst selection, synthesis procedure and the importance of the reactor set-up to maximize its utilization. Indeed, bibliography research has been carried out in order to compare similar photo-degradation processes with the attained results (Table 1). The term EC (Equation (3)) has been used to normalize the results, regardless of the initial MC concentration, the treatment time or the lamp power. Thus, the difference in EC is caused by either photocatalyst efficiency or reactor set-up.

In relation to photocatalyst suitability, the current results are superior to those of other authors, who have tried the addition of other compounds to TiO_2,_ such as SiO_2_ [1], V and Mo [6] or even natural zeolite [21]. This highlights the suitability of the combination of GO-TiO_2_, where GO addition favors the attraction between the drug and the photocatalyst, accelerating the degradation process [4] due to the acceleration of surface reactions [5]. This increased attraction is caused by the higher acidity and surface area when compared to raw TiO_2_, as it was demonstrated in Figure 4. Moreover, the GO addition increases the electron transfer capacities of GO-TiO_2_ [5], as it was demonstrated by the electrochemical impedance measurements (Figure 5).

Nevertheless, the efficiency has been improved when compared to other GO-TiO_2_-based photocatalysts (Table 1). In this case, the reactor set-up may be the differentiating aspect, as for instance, Wafi et al. [5] placed the lamp at 10 cm from the solution, when it is well known that the radiation intensity decreases with distance [2,22]. This fact has been proven when comparing the MC degradation by placing the wide-open reactor either at 2 or 6 cm from the UVA lamp (Figure 3). The results demonstrated more than 30% improvement when the lamp was placed closer. This is in concordance with the work of Sun et al. [22], who degraded 10% more MC within a film with their N-TiO_2_ photocatalyst when the lamp was placed at 0.5 cm from the film than when it was placed at 3 cm, caused by, in the latter case, lower photo current intensity due to the distance [2]. Moreover, the synthesis process was much easier in this presented study. Thus, Wafi et al.’s experiment [5] not only required calcination at 550 °C, but also the addition of dangerous reagents, such as NaBH_4_. Other authors’ experiments required long synthesis processes [4,23], instead of the brief GO-TiO_2_ process proposed in this work (Section 2.2).

Additionally, the EC of previous studies (Table 1) is much higher than the attained EC in this study, which demonstrates the efficiency of the reactor set-up. Indeed, it was not only the aforementioned reduction in the distance between the bulb solution and lamp that provided a much higher performance, but also the increase in the radiated surface area. In fact, this reality is also proven in Figure 3, where it can be noticed that increasing the irradiated area from 19.6 to 78.5 cm^2^ (for the narrow-open reactor and the wide-open reactor, respectively) caused an augmentation of around 40% in MC degradation. This was caused by an enhancement of the catalyst exposure to radiation and by the uniform irradiance of the photocatalyst through the small reactor depth [24].

Moreover, out of the selected references, no authors have validated their process with real wastewater effluents (Figure 8). Furthermore, only Mahata et al. [1] and Chakinala et al. [3] validated the reusability of the catalyst, although neither of them considered the utilization of an extra oxidant and even though Chakinala et al. [3] faced an almost 40% decrease in pollutant degradation. H_2_O_2_ addition may have increased the lifetime of their Bi-Ag-TiO_2_ photocatalyst.

## 5. Conclusions

GO-TiO_2_ was successfully synthesized and characterized in order to assess its performance as a dual (UV–Vis) photocatalyst. MC was used as a model pollutant in order to understand the photodegradation process. Thus, characterization and MC degradation unite to offer conclusions when compared to highly performant TiO_2_. For instance, GO-TiO_2_ resulted in a higher surface area, which experimentally implied more MC absorption under dark conditions, and a higher degradation rate due to better interactions of the drug-photocatalyst when compared to TiO_2_. Regarding the higher electron transfer of GO-TiO_2_, measured by Nyquist graphs, it was clear that the electron–hole recombination was diminished in contrast to TiO_2_, which also explained the better photo-activity of the former. Moreover, the results were validated by reusing the several photocatalyst cycles by the assistance of H_2_O_2_ addition, by treating real wastewaters and by the comparison to previous data. Indeed, the EC was 0.12 Wh/mg, which has not been achieved previously; this makes room for further studies regarding the usage of the GO-TiO_2_ catalyst and similar reactor set-ups.

## Figures and Tables

**Figure 1 ijerph-19-11874-f001:**
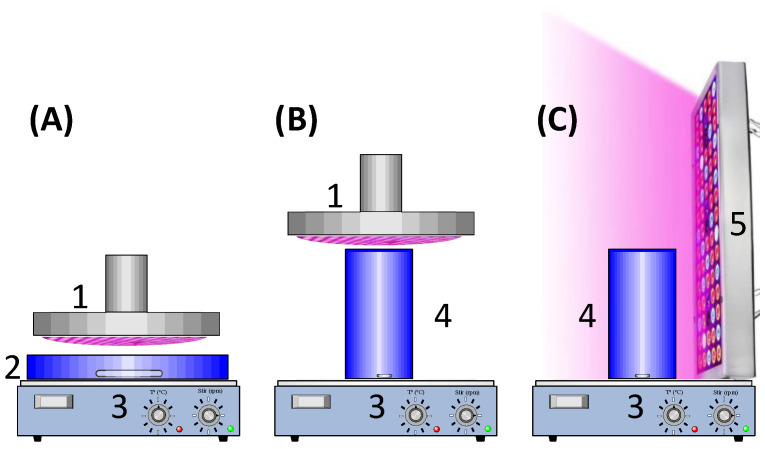
Proposed reactors: (**A**) wide-open-UVA reactor, (**B**) narrow-open-UVA reactor, (**C**) visible reactor, where 1: UVA LED lamp, 2: 10 cm ø, 1.5 cm high cylindrical reactor, 3: magnetic stirring, 4: 6 cm ø, 10 cm high cylindrical reactor, 5: visible lamp.

**Figure 2 ijerph-19-11874-f002:**
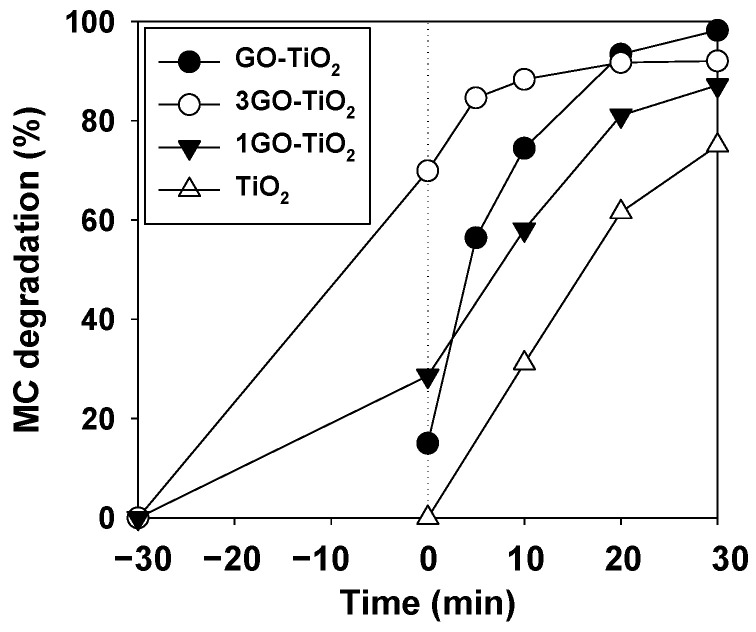
MC photodegradation (20 mg/L) at catalyst concentration of 800 mg/L under UV radiation. TiO_2_ (white triangles), ½GO-TiO_2_ (so-called GO-TiO_2_) (black circles), 1GO-TiO_2_ (black triangles), and 3GO-TiO_2_ (white circles).

**Figure 3 ijerph-19-11874-f003:**
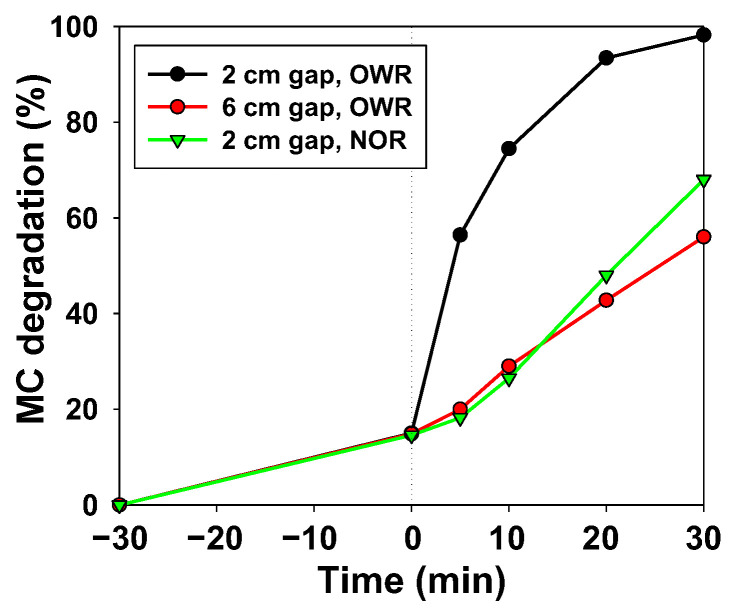
MB degradation on the open-wide-reactor (OWR) and narrow-open-reactor (NOR), placing the UV-lamp at different gaps.

**Figure 4 ijerph-19-11874-f004:**
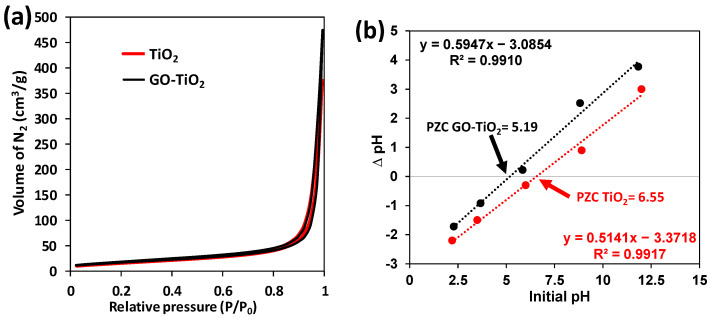
Mechanisms explaining the absorption behavior: (**a**) N_2_ isotherm and (**b**) point of zero charge of TiO_2_ and GO-TiO_2_.

**Figure 5 ijerph-19-11874-f005:**
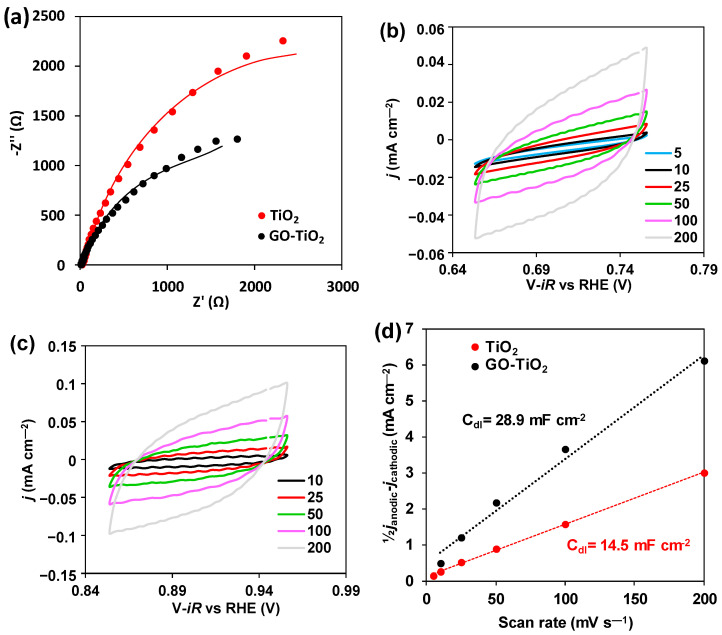
Electrochemical measurements: (**a**) Nyquist graph, where the lines represent the equivalent circuit adjustment; (**b**,**c**) are CVs at different scan rates (in mV s^−1^) for TiO_2_ (**b**) and TiO_2_-GO (**c**). CDL representation of both samples (**d**).

**Figure 6 ijerph-19-11874-f006:**
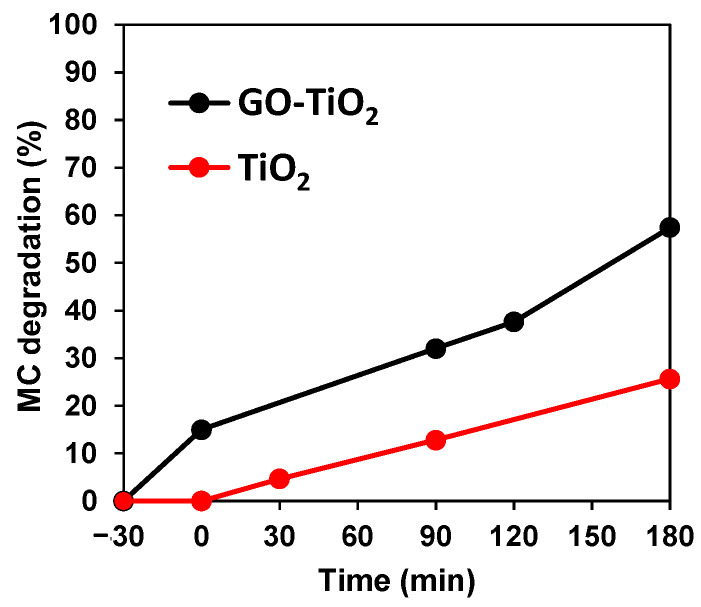
MC photodegradation (20 mg/L) at the catalyst concentration of 800 mg/L with TiO_2_ and GO-TiO_2_ under simulated solar radiation.

**Figure 7 ijerph-19-11874-f007:**
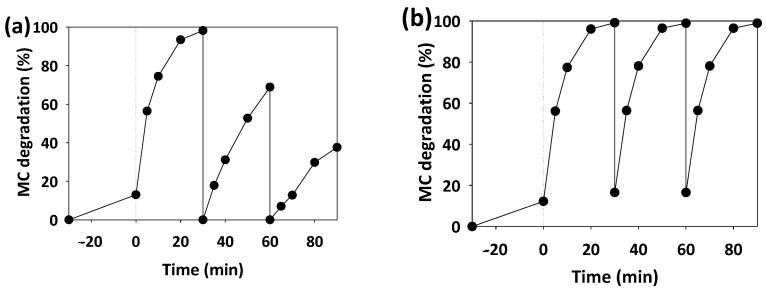
MC photodegradation under UV radiation (20 mg/L each cycle), using the same photocatalyst (at a concentration of 800 mg/L) during three cycles (**a**) and with the addition of 0.66 mg/mL of H_2_O_2_ in each cycle (**b**).

**Figure 8 ijerph-19-11874-f008:**
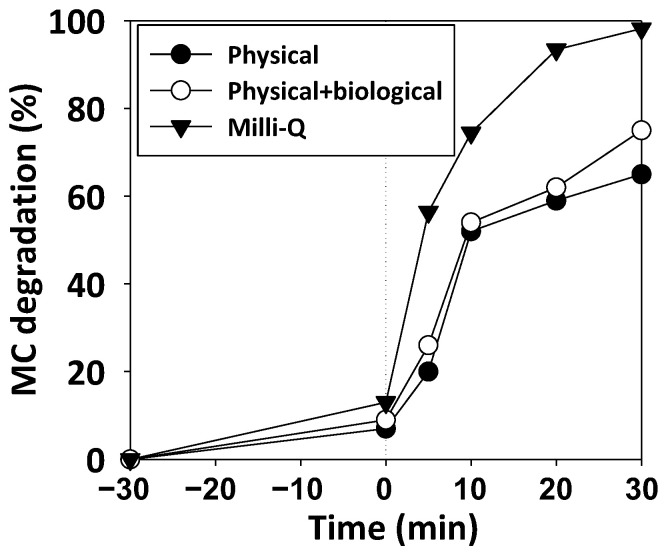
MC photodegradation (20 mg/L) at catalyst concentration of 800 mg/L under UV radiation. Matrix composed of real wastewater that received a physical treatment (black circles), matrix that received a physical and biological treatment (white circles), and milli-Q water (black triangles).

**Table 1 ijerph-19-11874-t001:** Recent studies on MC degradation with UVA photocatalysis.

MC (mg/L)	Catalyst (mg/L)	Lamp (nm, w)	Time (min)	Degradation (%)	EC (W·h/mg)	Reference
20	GO-TiO_2_ (800)	360–365 nm, 4.8 W	30	100	0.12	This study
10	TiO_2_-SiO_2_ (1000)	8 W	30	85	0.47	[1]
10	TiO_2_ nanoparticles (60)	UV, 300 W, >420 nm	24	99	12.12	[2]
10	Bi-Ag-TiO_2_ (15)	Vis, 250 W	120	90	111.1	[3]
5	GO-TiO_2_ (200)	500 W	60	92	108.70	[4]
6.4	Reduced GO-TiO_2_ (1000)	6 W, 365 nm	60	86	1.09	[5]
3.2	V/Mo-TiO_2_ (1000)	365 nm, 8 W	60	86.7	2.88	[6]
10	Zeolite/TiO_2_ (50)	UV, 16 W	120	100	3.20	[21]
73.57	Graphene-TiO_2_ (50)	360 nm, 17 W	480	87	2.12	[23]

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
