# Peer review of "GO-TiO2 as a Highly Performant Photocatalyst Maximized by Proper Parameters Selection"

_ijerph, 2022, doi:10.3390/ijerph191911874_

Round 1
Reviewer 1 Report
The study shows TiO2 catalyst combined with graphene oxide and testing for UV and Vis degradation of organic pollutant MC.
The manuscript and study in general is well described. Improvements that must be completed are:
1. English errors in the opening paragraphs (for example line 28 "were" - "where")
2. Some important figures from the SI should be included in the main manuscript. For example: Figures 2SM, 3SM, 7SM, and 8SM are critical to the manuscript and should be placed in the main manuscript, and not in the supporting info. If appropriate, ones that relate to each other can be put in multi-panel figures.
Author Response
Dear Reviewer,
We are pleased to enclose the revised version of our manuscript entitled “GO-TiO2 as a well-performant photocatalyst maximized by a proper parameters selection” for its evaluation for the publication on the International Journal of Environmental Research and Public Health. We strongly believe that the quality of the manuscript has been enhanced by following the reviewer’s comments attained after submission number ijerph-1835878 where the article was entitled “Graphene oxide as an efficient doping for enhancing P25 photocatalyst performance”.
- English errors in the opening paragraphs (for example line 28 "were" - "where")
Response to the comment: Thanks to the reviewer’s comment, we have checked the English carefully
- Some important figures from the SI should be included in the main manuscript. For example: Figures 2SM, 3SM, 7SM, and 8SM are critical to the manuscript and should be placed in the main manuscript, and not in the supporting info. If appropriate, ones that relate to each other can be put in multi-panel figures.
Response to the comment: regarding the reviewer’s suggestion, we have included some important figures on the manuscript.
Reviewer 2 Report
The manuscript of Aida M. Díez et al. “Graphene oxide as an efficient doping for enhancing P25 photocatalyst performance” describes the modification of TiO2 photocatalyst powder with graphene oxide (GO) to improve the efficiency of methylthioninium (MC) photocatalytic degradation, which served as a model drug.
Major improvements are required before acceptance:
1. The title should be improved with more details such as “Enhanced photocatalytic performance of GO-TiO2 synthesized by hydrothermal method and subsequent heating”.
2. Specify in 2.1. Reagents the use of commercial TiO2 Degussa P25 for comparison with synthesized GO-TiO2 and add manufacturer specifications, such as phase content, particle size and shape, surface properties, or add references.
3. Add SEM images of GO-TiO2 samples for morphology and particle size characterization, if possible.
4. There are many typos and grammatical errors in the paper, please proofread the paper carefully. Check for typos such as the underlined temperature degree on page 2, lines 70 and 71; FITR on page 2, line 80; N with 2 in superscript on page 2, line 81; experience, page 2, lines 93-94, etc. Check for style errors such as the sentence on page 2, line 74, or the sentence “The slope provided after linear fitting of this graph provides…” on page 3, lines 98-99, and throughout.
5. When referring to synthesized samples, or in the case of commercial TiO2, you must always use the same labels after the first definition of those, e.g. use of the same abbreviations for samples throughout the text. For example, the use of GO-TiO2 (10) on page 3, line 135; or TiO2-GO on page 2, lines 71-72 is not suitable.
6. Add a sentence at the end of page 3 regarding the reason for choosing GO-TiO2 (0.1) as the best candidate, which is its lowest adsorption affinity to MC, although synergistic adsorption-photocatalytic degradation of MC is considered the best for GO-TiO2 (0.05) in 30 min (Fig. 1).
7. Indicate in the descriptions of Fig. 1, Fig. 4, and Fig. 8-SM that the degradation of MC was carried out under UV irradiation. Write in Fig. 3-SM the concentration of MC and the catalyst loading.
8. Add a legend to Fig. 1 and Fig. 8-SM, and TiO2 Degussa P25 and GO-TiO2 (0.1) marks in Fig. 3 b) and c), respectively.
9. Plot XRD patterns (Fig. 4-SM) one below the other, add reference JCPDS cards for anatase and rutile phases used to identify their peaks, and index with (hkl) each peak.
10. Assign the vibration modes to the FTIR peaks in Fig. 5-SM.
11. Check the correctness of the paragraph on page 6, lines 192 to 194, and add the correct data (%) of MC degradation under Vis irradiation shown in Figure 7-SM.
12. Change the term “MC elimination” to “MC degradation” or “MC photodegradation” throughout the text, as well as in the Figures.
Author Response
Dear Reviewer,
We are pleased to enclose the revised version of our manuscript entitled “GO-TiO2 as a well-performant photocatalyst maximized by a proper parameters selection” for its evaluation for the publication on the International Journal of Environmental Research and Public Health. We strongly believe that the quality of the manuscript has been enhanced by following the reviewer’s comments attained after submission number ijerph-1835878 where the article was entitled “Graphene oxide as an efficient doping for enhancing P25 photocatalyst performance”.
Major improvements are required before acceptance:
- The title should be improved with more details such as “Enhanced photocatalytic performance of GO-TiO2synthesized by hydrothermal method and subsequent heating”.
Response to the comment: Following the appropriate reviewer´s comment, we have modified the article´s title for “GO-TiO2 as a well-performant photocatalyst maximized by proper parameters selection”, taking into account the reviewer asked for a complete title, and we thought that it is more important in our paper the fact we studied different parameters rather than the synthesis process on GO-TiO2.
- Specify in 2.1. Reagents the use of commercial TiO2Degussa P25 for comparison with synthesized GO-TiO2and add manufacturer specifications, such as phase content, particle size and shape, surface properties, or add references.
Response to the comment: Thanks to the reviewer’s comment, we have added such information on the manuscript (on the section reagents) and moreover we would like to highlight that the structural properties are depicted on table 2-SM.
- Add SEM images of GO-TiO2 samples for morphology and particle size characterization, if possible.
Response to the comment: We are grateful to the reviewer for its comment, nevertheless, we have already lots of figures and we have provided data ons trucutre thanks to N2 isotherm, XRD and FTIR analysis.
- There are many typos and grammatical errors in the paper, please proofread the paper carefully. Check for typos such as the underlined temperature degree on page 2, lines 70 and 71; FITR on page 2, line 80; N with 2 in superscript on page 2, line 81; experience, page 2, lines 93-94, etc. Check for style errors such as the sentence on page 2, line 74, or the sentence “The slope provided after linear fitting of this graph provides…” on page 3, lines 98-99, and throughout.
Response to the comment: We are sorry for those errors, and we have corrected them and checked carefully the whole manuscript.
- When referring to synthesized samples, or in the case of commercial TiO2, you must always use the same labels after the first definition of those, e.g. use of the same abbreviations for samples throughout the text. For example, the use of GO-TiO2(10) on page 3, line 135; or TiO2-GO on page 2, lines 71-72 is not suitable.
Response to the comment: We regret this mistake and we have corrected the labeling of the samples thanks to the reviewer’s comment
- Add a sentence at the end of page 3 regarding the reason for choosing GO-TiO2(0.1) as the best candidate, which is its lowest adsorption affinity to MC, although synergistic adsorption-photocatalytic degradation of MC is considered the best for GO-TiO2 (0.05) in 30 min (Fig. 1).
Response to the comment: Thanks to the reviewer’s comment, we have noticed the mistake on the labelling, actually, we have selected Go-TiO2 (0.05) as the best alternative and we hope now it is clear thanks to the labeling changes (so-called ½GO-TiO2)
- Indicate in the descriptions of Fig. 1, Fig. 4, and Fig. 8-SM that the degradation of MC was carried out under UV irradiation. Write in Fig. 3-SM the concentration of MC and the catalyst loading.
Response to the comment: Thanks to the reviewer’s comment, we have added that info on the graphs
- Add a legend to Fig. 1 and Fig. 8-SM, and TiO2Degussa P25 and GO-TiO2 (0.1) marks in Fig. 3 b) and c), respectively.
Response to the comment: Thanks to the reviewer’s comment, we have added that info on the graphs
- Plot XRD patterns (Fig. 4-SM) one below the other, add reference JCPDS cards for anatase and rutile phases used to identify their peaks, and index with (hkl) each peak.
Response to the comment: Thanks to the reviewer’s comment, we have added that info on the graphs
- Assign the vibration modes to the FTIR peaks in Fig. 5-SM.
Response to the comment: Thanks to the reviewer’s comment, we have added that info on the graphs
- Check the correctness of the paragraph on page 6, lines 192 to 194, and add the correct data (%) of MC degradation under Vis irradiation shown in Figure 7-SM.
Response to the comment: Thanks to the reviewer’s suggestion, we have corrected the data on the manuscript.
- Change the term “MC elimination” to “MC degradation” or “MC photodegradation” throughout the text, as well as in the Figures.
Response to the comment: Following the reviewer’s suggestion, we have changed the terms, included on the graphs axis
Reviewer 3 Report
Dear authors. I enjoyed reading your manuscript and I have some comments and suggestions. Good luck
English language needs significant improvement.
Electron microscopic studies are needed to understand the role of GO in the catalytic system. It is important to understand how size and morphology effects the activity.
XPS studies will significantly improve the quality of the manuscript as it will show whether GO is in the bulk or on the surface. Further comparison of XPS of used samples with the ones treated with H2O2 can give important information on catalyst reactivation.
On the minor side XRD is not a spectra. It is a diffraction study.
Author Response
Dear Reviewer,
We are pleased to enclose the revised version of our manuscript entitled “GO-TiO2 as a well-performant photocatalyst maximized by a proper parameters selection” for its evaluation for the publication on the International Journal of Environmental Research and Public Health. We strongly believe that the quality of the manuscript has been enhanced by following the reviewer’s comments attained after submission number ijerph-1835878 where the article was entitled “Graphene oxide as an efficient doping for enhancing P25 photocatalyst performance”.
Dear authors. I enjoyed reading your manuscript and I have some comments and suggestions. Good luck
- English language needs significant improvement.
Response to the comment: Thanks to the reviewer’s suggestion, we have improved the English quality of the manuscript.
- Electron microscopic studies are needed to understand the role of GO in the catalytic system. It is important to understand how size and morphology effects the activity.
Response to the comment: We agree with the reviewer’s comment. Nevertheless, this is a preliminary study where we focus on the GO selection, reactor set-up and initial characterization. In following studies, we are going to focus on the synthesis process so depending on temperature, pressure and dissolvent type we may attain different structures which would be analyzed.
- XPS studies will significantly improve the quality of the manuscript as it will show whether GO is in the bulk or on the surface. Further comparison of XPS of used samples with the ones treated with H2O2can give important information on catalyst reactivation.
Response to the comment: Thanks to the reviewer’s comment. We have already plenty of graphs and it has been reported GO-TiO2 is usually placed on the surface and we have added this information to the manuscript and it is in concordance with FTIR and XPR results, where the spectra is almost equal, demonstrating an invariable structure.
- On the minor side XRD is not a spectra. It is a diffraction study.
Response to the comment: We have corrected this mistake thanks to the reviewer’s comment.
Reviewer 4 Report
The authors presented TiO2 photocatalysts doped with graphene oxide (GO). There are several issues that the authors need to address:
1. The most serious issue seems to be that the GO-TiO2 does not seem to lead to better MC elimination under visible light, compared to pure TiO2 (Fig 7 SM). This seems to invalidate the main conclusions of this paper.
2. In the introduction, the authors cited several different dopants for TiO2, but did not motivate the use of GO. They should explain why they chose to use GO, and also cite previous GO literature.
3. From the UV-Vis spectra, it is hard to say whether the TiO2-GO composite absorbs more visible light than pure TiO2. The authors need to do a proper Tauc plot for band gap extraction.
4. Series resistance should also be extracted from the impedance data, in addition to capacitance.
5. It is puzzling to me why the authors choose to place the bulk of their data in the supplementary material. Important data such as XRD, FTIR, UV-Vis and also degradation results should be incorporated into the main text.
6. Suggestion for English edit: over-use of the word "attained". In some instances, it should be "purchased from", in others, it should be "obtained".
Author Response
Dear Reviewer,
We are pleased to enclose the revised version of our manuscript entitled “GO-TiO2 as a well-performant photocatalyst maximized by a proper parameters selection” for its evaluation for the publication on the International Journal of Environmental Research and Public Health. We strongly believe that the quality of the manuscript has been enhanced by following the reviewer’s comments attained after submission number ijerph-1835878 where the article was entitled “Graphene oxide as an efficient doping for enhancing P25 photocatalyst performance”.
The authors presented TiO2 photocatalysts doped with graphene oxide (GO). There are several issues that the authors need to address:
- The most serious issue seems to be that the GO-TiO2 does not seem to lead to better MC elimination under visible light, compared to pure TiO2 (Fig 7 SM). This seems to invalidate the main conclusions of this paper.
Response to the comment: We in fact present data now on figure 6 and we can see how the MC degradation is twice with GO-TiO2 than with TiO2, so, as a prelyminary test of dual activity, we can conclude that our catalyst exhibits some activity. Nevertheless, the article is focused on UVA degradation, this is why the whole remaining manuscript present UVA results. We only add visible data as an option for further research and in order to highlight the catalyst has several advantages.
- In the introduction, the authors cited several different dopants for TiO2, but did not motivate the use of GO. They should explain why they chose to use GO, and also cite previous GO literature.
Response to the comment: We are grateful to the reviewer for the comment. We have added the fact that more authors have added GO to TiO2 but the efficiencies have not been quite good maybe due to reactor set-up (fact discussed more deeply on the discusion section)
- From the UV-Vis spectra, it is hard to say whether the TiO2-GO composite absorbs more visible light than pure TiO2. The authors need to do a proper Tauc plot for band gap extraction.
Response to the comment: Thanks to the reviewer for the suggestion, as it can be seen on figure 4-SM the new graph with Tauc ploc included, and we have added the band gap exact value on the manuscript
- Series resistance should also be extracted from the impedance data, in addition to capacitance.
Response to the comment: We have added the equivalent circuits on Fig. 5 of Supplementary material and we have commented the attained data on the manuscript on section 4.2.8
- It is puzzling to me why the authors choose to place the bulk of their data in the supplementary material. Important data such as XRD, FTIR, UV-Vis and also degradation results should be incorporated into the main text.
Response to the comment: We are grateful to the reviewer for the suggestion, and consequently we have moved that data to the main text
- Suggestion for English edit: over-use of the word "attained". In some instances, it should be "purchased from", in others, it should be "obtained".
Response to the comment: We are very grateful to the reviewer’s comment, so we have noticed this inmense repetition of the word ”attained”. Now we have changed it on the whole manuscript for synonims such as accomplished, got, achieved or obtained.
Round 2
Reviewer 2 Report
There are still typos to be corrected, like already sugested: "FITR" first word on page 2, line 86; "N with 2 in superscript" on page 2, line 87, that the authors fail to correct again. Chech for others.
Regarding my previous requests:
- Plot XRD patterns (Fig. 4-SM) one below the other, add reference JCPDS cards for anatase and rutile phases used to identify their peaks, and index with (hkl) each peak.
Response to the comment: Thanks to the reviewer’s comment, we have added that info on the graphs
AND,
- Assign the vibration modes to the FTIR peaks in Fig. 5-SM.
Response to the comment: Thanks to the reviewer’s comment, we have added that info on the graphs
I dont see in the revised manuscript that you did actualy add the requested info on the graphs (in revised manuscript Fig. 2-SM and Fig. 3-SM)!
Author Response
We are gratefull to the reviewer for the comments. We have checked again the manuscript and correct some typos. Moreover, we have addeed XRD and FTIR new graphs (which we are deeply sorry to have forgot).
Reviewer 4 Report
The manuscript has improved after revision, and I am happy to recommend it for publication,
Author Response
We are very gratefull to the reviewer for the possitive feedback, and we have read the manuscript carefully to correct some minor errors thanks to the reviewer's suggestion